# Hedgehog Signalling Modulates Immune Response and Protects against Experimental Autoimmune Encephalomyelitis

**DOI:** 10.3390/ijms23063171

**Published:** 2022-03-15

**Authors:** Alicia Ballester, Adriana Guijarro, Beatriz Bravo, Javier Hernández, Rodolfo Murillas, Marta I. Gallego, Sara Ballester

**Affiliations:** 1Unidad de Regulación Génica, Unidad Funcional de Investigación en Enfermedades Crónicas, Instituto de Salud Carlos III, Carretera Majadahonda-Pozuelo Km2, 28220 Madrid, Spain; alicia.ballester@isciii.es (A.B.); aguijarro@isciii.es (A.G.); b.bravo@cajal.csic.es (B.B.); javiht@isciii.es (J.H.); 2Unidad de Innovación Biomédica, Centro de Investigaciones Energéticas, Medioambientales y Tecnológicas (CIEMAT), Avenida Complutense, 40, 28040 Madrid, Spain; rodolfo.murillas@ciemat.es; 3Unidad de Histología y Patología mamaria, Instituto de Salud Carlos III, Carretera Majadahonda-Pozuelo Km2, 28220 Madrid, Spain

**Keywords:** Hedgehog, Patched-1, Th lymphocytes, experimental autoimmune encephalomyelitis, multiple sclerosis

## Abstract

The Hedgehog (Hh) pathway is essential for the embryonic development and homeostatic maintenance of many adult tissues and organs. It has also been associated with some functions of the innate and adaptive immune system. However, its involvement in the immune response has not been well determined. Here we study the role of Hh signalling in the modulation of the immune response by using the Ptch-1-LacZ^+/−^ mouse model (hereinafter referred to as *ptch*^+/−^), in which the hemizygous inactivation of Patched-1, the Hh receptor gene, causes the constitutive activation of Hh response genes. The in vitro TCR stimulation of spleen and lymph node (LN) T cells showed increased levels of Th2 cytokines (IL-4 and IL-10) in *ptch*^+/−^cells compared to control cells from wild-type (wt) littermates, suggesting that the Th2 phenotype is favoured by Hh pathway activation. In addition, CD4^+^ cells secreted less IL-17, and the establishment of the Th1 phenotype was impaired in *ptch*^+/−^ mice. Consistently, in response to an inflammatory challenge by the induction of experimental autoimmune encephalomyelitis (EAE), *ptch*^+/−^ mice showed milder clinical scores and more minor spinal cord damage than wt mice. These results demonstrate a role for the Hh/ptch pathway in immune response modulation and highlight the usefulness of the *ptch*^+/−^ mouse model for the study of T-cell-mediated diseases and for the search for new therapeutic strategies in inflammatory diseases.

## 1. Introduction

Hedgehog (Hh) signalling is an evolutionally conserved pathway that determines cell fate during organogenesis and plays fundamental roles in the embryonic patterning of vertebrates and invertebrates [1]. This pathway is involved in cell fate in different organs and tissues, including bones, gonads, the intestine, the heart, lungs and muscles. The role of Hh in determining cell fate in the central nervous system is well documented, contributing to the morphogenesis and functionality of the cerebellum, eye, neural crest, peripheral nerves, neurons and oligodendrocytes [2]. However, although the participation of Hh in the differentiation of T cells in the thymus has been described, its role in immune regulation has not yet been fully elucidated [3,4].

The main components of the Hh pathway are its three different secreted protein ligands Sonic, Dessert and Indian; Patched (Ptch), its transmembrane receptor; Smoothened (Smo), a G-protein-coupled receptor that carries the Hh signal across the cell membrane; and the three Gli transcription factors. In the absence of a ligand, Ptch constitutively represses the activity of Smo, causing Gli protein transcription activity to remain repressed by protein complexes, including Sufu, PKA, GSK3 and CK1, among other factors. The binding of Hh ligands to the Ptch receptor alleviates the inhibition of Smo, which results in the release of Gli proteins and their transfer to the nucleus where they activate the transcription of the target genes of the pathway, among which is the Gli1 gene itself [5]. Gli1, in turn, transcribes more Hh target genes in a positive loop of signalling. Another target gene of the pathway is Ptch itself, which, by being able to act again as an Smo repressor, acts as a negative self-regulator [6]. This sequence constitutes the canonical Hh signalling, although non-canonical pathways have also been described in which Hh signalling is independent of Gli proteins [7].

Aberrant Hh signalling not only produces malformations during development but has also been associated with tumour progression in adulthood. Inhibitors of the Hh pathway reduce the proliferation of tumour cells and prevent metastasis in different types of cancer [8,9,10]. The targeted inhibition of Smo and Gli1 activity has provided information on the role of the Hh/Ptch pathway in the immune response against tumour cells. Thus, Hh signalling influences the activity of macrophages in the tumour microenvironment [11,12,13], as well as the activation of T lymphocytes [3,4,14,15] and the polarisation of T cells towards different T helper (Th) cell phenotypes (Th1, Th2 and Th17) [16,17,18,19] and modulates the development of different inflammatory pathologies, such as arthritis, pancreatitis, dermatitis, intestinal inflammations or experimental autoimmune encephalomyelitis (EAE) [18,20,21,22,23,24,25].

EAE is a model of multiple sclerosis (MS), promoted by autoreactive Th1 and Th17 cells that are activated in peripheral immune organs, such as the lymph nodes (LN) and spleen, and migrate to the central nervous system (CNS) where, together with inflammatory macrophages, trigger the CNS’s neuropathological response, characterised by chronic inflammation leading to demyelination. The inflammatory cascade involves the expansion of reactive microglia and astrocytes. This is particularly notable in the highly infiltrated spinal cord, where the increase in the number of astrocytes and microglia cells correlates with the severity of EAE [26].

In this report, we used the hemizygous Ptch-1-LacZ mice (*ptch*^+/−^) generated by Goodrich et al. [27] in which one of the copies of the ptch-1 gene is inactivated by substituting exon 2 with the neomycin resistance and LacZ genes. The deficiency in the protein ptch-1 causes the activation of the Hh pathway by not repressing its constitutive activation by Smo. This model has been repeatedly used to investigate the effects of Hh pathway over-activation on developments and tumourigenesis [28]. The objective of our study was to evaluate the role of the Hh pathway in the modulation of the immune response by the use of *ptch*^+/−^ and control mice. We have analysed the cytokine profiles of T cells and macrophages in cultures of *ptch*^+/−^ and wild-type (wt) cells in the absence of an inflammatory pathology. We explored the consequences of an inflammatory challenge, such as EAE, by using the evaluation of a clinical score and by performing a histological and immuno-histochemical characterisation of spinal cord lesions. Our results demonstrate that the Hh/ptch pathway plays a role in modulating the immune response and that the overactivation of the Hh pathway plays a protective role against EAE.

## 2. Results

### 2.1. The Th2 Profile Is Favoured in T lymphocytes from ptch^+/−^ Mice

In order to characterise the T helper (Th) profile of T lymphocytes in *ptch*^+/−^ mice, splenocytes and inguinal and axillary lymph node cells (LNC) were isolated from wild-type (wt) and heterozygous animals. Whole-cell populations were cultured in the presence of plate-bound anti-CD3 and soluble anti-CD28 (αCD3/αCD28) to stimulate T lymphocytes through the TCR complex. After three days of a culture, the supernatants were assayed by using an ELISA for interleukins. Cells from *ptch*^+/−^ mice from both the spleen and LN produced significantly higher amounts of IL-4 and IL-10 compared to wt cells (Figure 1), two typical cytokines produced by Th2 cells. Spleen-purified *ptch*^+/−^ CD4^+^ cells also showed higher levels of IL-4 and IL-10 than their wt counterparts, indicating that the bias towards the Th2 phenotype is not due to the influence of other cell types present in the cultures.

To further characterise the Th polarisation in *ptch*^+/−^ mice, we compared the production of IFNγ and IL-17 as distinctive cytokines produced by Th1 and Th17 phenotypes, respectively, in cultures of LN and spleen cells as well as in cultures of purified CD4^+^ cells from *ptch*^+/−^ and wt mice. Although purified CD4^+^ cells from *ptch*^+/−^ spleens secreted less IL-17 than CD4^+^ wt cells, as expected in a Th2-favored phenotype, no differences between genotypes were found when the total populations of splenocytes or LNCs were compared. This suggests that in these heterogeneous cell cultures, the IL-17 production by CD4^+^ cells could be influenced by other cell types present (Figure 2A). However, in LNC populations, IFNγ production did not show significant differences between both genotypes; in complete spleen populations and in purified CD4^+^ cells, it was higher in *ptch*^+/−^ than in wt mice (Figure 2B). Furthermore, under Th17 and Th1 skewing conditions, the production of IL-17 and IFNγ, respectively, was similar for both genotypes (Figure 2A,B).

We next analysed how *ptch*^+/−^ T lymphocytes respond to sustained TCR stimulation. Spleen cells were subjected to a second round of αCD3/αCD28 stimulation three days after the first challenge. As found for the first round of TCR stimulation, the secreted amounts and mRNA levels of IL-4 and IL-10 were higher in the ptch mutant cells than in wt cells (Figure 3A,B). The IL-17 quantification showed no differences between *ptch*^+/−^ and wt mice in the complete spleen nor in LN cell (LNC) populations, but purified CD4^+^ cells from *ptch*^+/−^ secreted less IL-17 than CD4^+^ wt cells, the same as in the previous experiment with a single TCR stimulation (Figure 3C). Both mouse genotypes showed similar levels of IFNγ production, unlike that observed after a single TCR stimulation. Furthermore, under an induction of the Th1 phenotype by using IL-12, the T cells from *ptch*^+/−^ mice responded to a second round of TCR activation, secreting a lower IFNγ production than wt cells (Figure 3D).

These data on cytokine profiles suggest an unbalanced T helper cell polarisation under TCR stimulation in *ptch*^+/−^ mice.

### 2.2. Peritoneal Macrophages Are Prone to M1 Phenotype

To characterise *ptch*^+/−^ macrophages, we compared the expressions of several macrophage activity markers in wt and *ptch*^+/−^ mice. Peritoneal macrophages were activated in vitro by using an LPS stimulation, and the expression levels of the macrophage activity markers IL-6, TNFα, IFNγ and IL-1β e iNOS were quantified. The *ptch*^+/−^ macrophages showed a faster induction of all five markers than the wt macrophages, with significant differences after 4 h of culture. However, for four of these five markers, the differences in the mRNA levels between the *ptch*^+/−^ and wt macrophages were minor or non-existent for longer periods of time (24 h). iNOS was the exception, maintaining the differences in expression levels between *ptch*^+/−^ and wt cells over time (Figure 4).

Two different kinetic patterns were found in *ptch*^+/−^ macrophage activity markers. While IL-6, TNFα and IFNγ briefly increased quickly after the LPS treatment, followed by a reduction in wt levels during the following hours, IL-1 and iNOS expressions did not decrease in the *ptch*^+/−^ macrophages after twenty-four hours in a culture (Appendix A). These five markers are related to the inflammatory activity of macrophages of the M1 phenotype. To ascertain whether this tendency towards a rapid M1 response was associated with a lower expression of the M2 phenotype in *ptch*^+/−^ macrophages, peritoneal macrophages were cultured under M2-skewing conditions and the expression of the M2 distinctive marker Arg-1 was quantified. This marker was shown to be lower in *ptch*^+/−^ cells than in wt cells (Figure 4F), consistent with the observation of a sharper M1-type polarisation in *ptch*^+/−^ macrophages than in wt macrophages.

Since in *ptch*^+/−^, CD4^+^ cells produce less IL-17 than wt cells and the stabilisation of the Th1 phenotype is compromised (Figure 3), whereas for splenocyte cultures, it has been found there are higher IFNγ and IL17 expressions than expected (Figure 2), we wanted to find out whether the M1 profiles of *ptch*^+/−^ macrophages could contribute directly or indirectly to the expressions. Therefore, we repeated TCR activation experiments in total spleen cells but depleted the CD11b^+^ fraction prior to the cell culture and cytokine quantification. ELISA experiments showed markedly lower amounts of IFNγ and IL17 in the CD11b^+^-depleted spleen populations than in the samples of total-cell populations. However, the IFNγ and IL17 expression ratios between the *ptch*^+/−^ and wt populations did not change with respect to populations in which the CD11b^+^ fraction had not been eliminated (Figure 5). This demonstrates that the CD11b^+^ cells contribute to the total levels of IFNγ and IL-17, probably due to co-stimulatory interactions with T cells. However, this result rules out macrophages as the causes of higher IFNγ and IL17 expressions than predicted in *ptch*^+/−^ total spleen-cell cultures.

### 2.3. EAE Development Is Attenuated in Ptch^+/−^ Mice

The protective role of the Th2 phenotype in EAE is well documented [26], whereas the inflammation triggered by M1 macrophages worsens the symptoms of EAE [29]. Thus, the bias towards the Th2 polarisation of *ptch*^+/−^ lymphocytes along with the predisposition to the M1 phenotype of *ptch*^+/−^ peritoneal macrophages made it difficult to predict what the course of an inflammatory disease, such as EAE, would be like in *ptch*^+/−^ mice compared to that in wt mice. Statistical analyses of two large-scale EAE experiments demonstrated that *ptch*^+/−^ mice were protected from acute pathogenic hallmarks of autoimmune neuro-inflammation. The disease was shown to be milder in the *ptch*^+/−^ mice than in the wt mice. Although a few *ptch*^+/−^ animals showed severe symptoms, the average of the clinical score was lower for the *ptch*^+/−^ group than for the wt animals. Furthermore, the disease incidence evaluated thirty days after the inoculation of MOG was significantly reduced in the *ptch*^+/−^ group (Figure 6).

Histological and immuno-histochemical analyses of spinal cord cross-sections from *ptch*^+/−^ and wt mice, sacrificed on day 20 after an MOG inoculation, showed a profuse infiltration of immune cells in the periphery of the wt spinal cord sections in contrast to the uniform appearance of the spinal cord parenchyma of the *ptch*^+/−^ mice (Figure 7A). The outer parts of the spinal cords of the wt mice showed a vast disorganisation of the nervous tissue with a massive presence of immune cells compared to the intact nervous tissue in the *ptch*^+/−^ spinal cord (Figure 7B). The immunoreactivity of the infiltrating cells with the anti-CD11b antibody allowed for their characterisation as mainly myeloid cells, presented as round brown cells gathered on the outsides of the wt sections, while in the *ptch*^+/−^ samples, a uniform distribution of CD11b^+^ cells consistent with resident microglia was observed (Figure 7C). Microglial activation can be clearly detected with the anti-Iba1 antibody, which stained numerous amoeboid cells throughout the wt spinal cord parenchyma compared to scattered and thin branched microglia found in the spinal cord sections of the *ptch*^+/−^ mice (Figure 7D). The wt astrocytes exhibited more robust GFAP staining and a swollen appearance, both characteristic of reactive astrocytes, than the *ptch*^+/−^ astrocytes (Figure 7E). This higher reactivity observed in the wt spinal cord sections may be a consequence of the lesions caused by the infiltration of immune cells into their parenchyma, as shown in Figure 7E, but less frequent in *ptch*^+/−^ spinal cords. Stat3 phosphorylation has previously been associated with astrocyte activation [30]. To show differences in the astrocyte activation between the *ptch*^+/−^ and wt astrocytes after 20 days of an MOG35–55 inoculation, we performed a double GFAP/phospho-Stat3 immuno-histochemical analysis of *ptch*^+/−^ and wt spinal cord sections. Although phospho-Stat3 positive astrocytes were detected in wt and *ptch*^+/−^ mice in the outer parts of the spinal cord sections, the wt samples showed many more double-positive astrocytes for phospho-Stat3/GFAP (Figure 7F, thick arrows). Some infiltrating immune cells in the wt spinal cords also exhibited phospho-Stat3 staining in their nuclei, denoting cell activation (thin arrows). Polymorphic leukocyte nuclei were also distinguished within the infiltrated immune cells in the wt samples (asterisk). All of this histological information confirmed that the wt mice were in a more advanced EAE stage of the disease than the *ptch*^+/−^ mice at that time.

### 2.4. The Hh Pathway Is Activated in Glial Cells of ptch^+/−^ Mice Inoculated with MOG

We also investigated whether the low susceptibility to EAE symptoms of *ptch*^+/−^ mice correlates with deregulated Hh signalling in neural tissue. To detect the cells that respond to Hh signalling within the spinal cord, we performed an immuno-histochemical analysis with the anti-βgal antibody (a marker of the expression of ptch-1 in the *ptch*^+/−^ mouse model) in cross-sections from *ptch*^+/−^ mice inoculated with MOG_35–55_. We compared *ptch*^+/−^ animals without symptoms (score = 0) with some of the few with severe clinical scores (score = 4). Cells expressing Patched-1 were located primarily in the grey matter of spinal cord sections, and no significant changes in cell numbers or locations were found between the *ptch*^+/−^ mice with different degrees of responses to EAE induction (Figure 8A). Next, we investigated the nature of ptch-expressing glial cells through using a double immunofluorescence analysis of *ptch*^+/−^ spinal cord sections after inducing EAE (Figure 8B). The percentages of βgal^+^ cells co-expressing GFAP, Olig1 or NG2 are summarised in Figure 8C.

The double staining for βgal/Iba-1 showed that the βgal expressing cells were negative for Iba-1 expressions (Figure 8B), showing that the Hh pathway is not activated in microglial cells. The staining to detect βgal and the oligodendrocyte marker Olig1 showed that most oligodendrocytes which strongly expressed Olig1 did not co-express βgal. However, some βgal^+^ cells were faintly Olig1^+^, demonstrating that some cells in the oligodendrocyte lineage respond to Hedgehog signalling. This suggests that the Hh pathway is not activated in fully differentiated oligodendrocytes. To check if βgal^+^-positive oligodendrocytes were oligodendrocyte progenitor cells (OPCs), we double stained βgal and NG2, an OPC marker. A high percentage of the βgal^+^ cells were NG2^+^, supporting our hypothesis. In addition, all NG2-positive cells are also positive for the Olig1 marker, as shown in Supplementary Cells Figure 2. However, a double βgal/GFAP immunofluorescence revealed that the vast majority of βgal-expressing cells were astrocytes. This would imply that a fraction of the βgal^+^GFAP^+^ cells are also NG2^+^. Moreover, Olig2/GFAP (Appendix A) and Olig1/GFAP (Appendix A) double immuno-histochemistries revealed double positive cells in the *ptch*^+/−^ neural tissues, supporting the existence of a pool of progenitor cells expressing both oligodendrocyte and astrocyte lineage markers and able to respond to Hh signalling, which might play a role in spinal cord protection against EAE in *ptch*^+/−^ mice.

## 3. Discussion

We have examined the cytokine profiles of two main immune-cell types, T lymphocytes and macrophages, in the genetic context of Hh signalling upregulation using the Ptch-1 haplo-insufficiency Hh pathway activation model [27]. The activation of immune cells in this model of Hh overactivation shows an attenuated inflammatory response of Th cells, promoting polarisation towards Th2 in *ptch*^+/−^ (Figure 1 and Figure 3) cells, while a faster inflammatory M1 polarisation occurs in the *ptch*^+/−^ macrophages than in wt macrophages (Figure 4 and Appendix A).

Previous reports indicated that Gli2 activation modulates Th cell differentiation towards a Th2 phenotype [16,19,31] in correlation with our findings, which strongly suggests that the canonical Hh pathway plays a role in Th modulation. Our observation indicating that the activation of the Hh pathway promotes the establishment of the M1 phenotype is also consistent with previously published data showing that adipocyte-derived exosomes carrying Shh promoted the M1 polarisation of macrophages [32]. Moreover, Hh signalling has been linked to colitis-associated cancer by leading to M1 macrophage infiltration and the production of pro-inflammatory cytokines in a mouse model [33]. In addition, the administration of recombinant Shh increased the presence of IL-6 in the bone marrow stroma [34]. However, Hh signalling has also been associated with macrophage M2-type polarisation in other conditions. It has been reported that prolonged periods of inflammation in liver or lung tissues causes the secretion of Hh ligands that favour macrophage polarisation towards type M2 to control the inflammatory response [35]. Furthermore, a role for the M2 polarisation of infiltrating macrophages has been proposed for some types of tumours with the involvement of the Hh pathway [12,13,17]. The complexity of the regulation of the pathway and the sophisticated relationships among its components could be behind these divergences in different studies. Thus, the chosen strategy to manipulate the pathway, and the diversity of the physiological situations analysed, could also result in these apparently paradoxical results.

It is well known that the production of the distinctive cytokines by each Th subtype during their differentiation results in cross-inhibitory effects between each subtype with the others [36]. However, despite the higher production of Th2 cytokines, after the first TCR challenge in the total spleen or LN cell populations, neither IFNγ nor IL-17 production was lower in *ptch*^+/−^ cells than in wt cells (Figure 2). Nevertheless, *ptch*^+/−^ CD4^+^ cells presented deficient IL-17 production compared to wt CD4^+^ lymphocytes (Figure 2 and Figure 3). Among the M1 cytokines with faster inductions in *ptch*^+/−^ macrophages are IL-6 (a potent inducer of Th17 differentiation) and IFNγ (able to collaborate in Th1 differentiation), and they could be responsible for increasing the production of IL-17 and IFNγ in the total population of spleen cells (Figure 2). However, experiments in spleen-cell cultures in which the CD11b^+^ fraction was depleted rule out this possibility, since the depletion did not affect the IL17 and IFNγ ratios between the *ptch*^+/−^ and wt cells (Figure 5). To understand that the low levels of IL-17 produced by CD4^+^ cells from *ptch*^+/−^ cells are not reflected in the values obtained for the total spleen populations, it could be reasoned that the high IL-4 concentration in CD4^+^ cell cultures negatively regulates the production of IL-17. In the case of total spleen populations where only 10–20% of the total are CD4^+^ cells, the IL-4 levels might not be sufficient to impair IL-17 production. The data shown in Appendix A are consistent with this reasoning, with IL-4 and IL-17 concentrations being inversely correlated in the different types of cultures. However, in *ptch*^+/−^ cells primed with IL-12, a second round of TCR activation led to a reduced Th1 response measured as IFNγ production (Figure 3), suggesting that the high production of Th2 cytokines by *ptch*^+/−^ cells impairs the stabilisation of the Th1 phenotype and further supports a key role of the Hh pathway in attenuating the inflammatory activity of Th cells.

The immunopathology of EAE is governed by auto-reactive CD4^+^ cells, mainly Th1 and Th17 types, whereas the deviation to the Th2 phenotype has a protective effect [26]. Lymphocyte migration to the CNS triggers an inflammatory cascade that culminates in demyelination and neurodegeneration involving M1-profile macrophages. While macrophages and microglia M1 activity trigger the inflammatory process at the early phase of the disease, the M2 polarisation of both types of cells promotes the resolution of inflammation at later phases of the disease [29]. Polarised M1/M2 phenotypes are reversible depending on the cytokine environment, with IL-4 and IL-10 being important drivers towards M2 deviations. Here we show that *ptch*^+/−^ mice develop a more benign clinical form of EAE than the wt strain (Figure 6). The tendency of *ptch*^+/−^ mice towards Th2 lymphocytes in peripheral immune organs, compared to the wt genotype, possibly contributes to an overall result of less severe EAE. It is conceivable that the high production of IL-4 and IL-10 by *ptch*^+/−^ T lymphocytes could favour the M2 phenotype of infiltrating macrophages and resident microglia, which could collaborate to contain the inflammatory process more efficiently in *ptch*^+/−^ mice than in their wt littermates.

Histological and immuno-histochemical analyses of transverse spinal cord sections from *ptch*^+/−^ and wt mice, sacrificed on day 20 after the MOG inoculation, revealed the pathogenic hallmarks of autoimmune neuro-inflammation in wt sections, while the sections from most *ptch*^+/−^ animals were almost unaffected by EAE (Figure 7). The immune infiltration in the wt samples consisted mainly of myeloid cells gathered on the periphery of the wt sections. Stat3 activation in immune infiltrates has been shown to promote pathogenic myelin-specific T cell differentiation and leukocyte infiltration [37]. In agreement with this, we could distinguish a number of poly-morphonuclear cells near Stat3^+^-infiltrating cells in wt sections. The vast majority of EAE-induced *ptch*^+/−^ littermates did not show any spinal cord lesions nor any inflammatory infiltrations. Astrocyte immunofluorescence analyses, however, showed a tendency to show larger amounts of activated astrocytes in *ptch*^+/−^ samples than in wt asymptomatic and symptomatic samples (Appendix A). Even so, in the representative samples of the wt and *ptch*^+/−^ groups taken at day 20 after the MOG inoculation, the astrocyte activation features (swollen appearance and GFAP and Stat3 staining) were more prominent in the wt sections than in *ptch*^+/−^ sections (Figure 7E,F), consistent with a more advanced EAE stage than their *ptch*^+/−^ littermates.

Immuno-histochemistry and immunofluorescence analyses of β-gal^+^ cells, in combination with different glial makers, showed that most of the Hh-responding cells (β-gal^+^ cells) in the spinal cords of the *ptch*^+/−^ mice induced to EAE were astrocytes (Figure 8B) located in the grey matter (Figure 8A) regardless of disease stage. This is in agreement with previous reports showing that astrocytes are the predominant Hh-responding cells in the nervous system [38]. Similar to Ligon et al. [39], we also show that a high percentage of nuclei positive for β-gal^+^ nuclei localise in NG2^+^ OPCs, which in turn express Olig1 and Olig 2 markers (Appendix A). The percentages of βgal^+^GFAP^+^ and βgal^+^NG2^+^ cells were greater than 100% of the βgal^+^ cells (Figure 8C), which suggests that some βgal^+^NG2^+^ cells were also GFAP^+^. In fact, cells positive for both oligodendrocyte and astrocyte lineages could be detected in the *ptch*^+/−^ mice (Appendix A). Consistently, previous reports showed that NG2^+^ cells can be precursors of astrocytes and oligodendrocytes [40]. Furthermore, a role for the Hh/ptch pathway in the differentiation of NG2^+^ cells towards reactive astrocytes in a focal cerebral ischemia mouse model had been previously established [41]. It is conceivable that the higher amount of reactive astrocytes in *ptch*^+/−^ spinal cord sections than in wt spinal cord sections, shown in Appendix A, could be due to a greater drift from NG2^+^ cells towards reactive astrocytes promoted by overactivated Hh signalling in *ptch*^+/−^ mice. This higher basal level of reactivity-prone astrocytes could cope with neuroinflammation triggered by infiltrating immune cells more successfully in *ptch*^+/−^ animals than in their wt littermates and prevent excessive harmful astrogliosis and the onset of the disease. This strongly supports the proposal that Hh signalling ameliorates the course of EAE.

The involvement of the Hh pathway in mitigating neuroinflammation and axonal damage in MS and EAE has been previously reported, demonstrating that both genetic and pharmacological inactivations of Smo result in severe exacerbations of the clinical manifestations [20,42]. Various mechanisms were proposed that could mediate the positive influence of Hh signalling on EAE progression, possibly acting in parallel, such as neutralising the inflammatory activity of CD4^+^ cells or participating in the repair of the blood–brain barrier or in re-myelination processes [20,42,43]. Furthermore, it has been suggested that the Hh pathway participates in the therapeutic mechanisms of some known EAE treatments, such as Fingolimod or Niaspan [25,44].

We show here that genetically modified alleles causing the overactivation of Hh signalling resulted in a neuroprotection against EAE. Likewise, we demonstrate in the same model that increased Hh signalling predisposed CD4^+^ lymphocytes towards anti-inflammatory and neuroprotective Th2 polarisation. The modulation of the Th cell phenotype, together with the activity of the Hh pathway in the astrocytes and POCs of *ptch*^+/−^ animals, might collaborate to underlie EAE mitigation. In summary, our results show that the activity of the Hh pathway influences the immune response and its consequences in autoimmune and inflammatory diseases and reveals that the *ptch*^+/−^ mouse can be a useful model to use to identify novel therapeutic approaches for T-cell-mediated diseases.

## 4. Materials and Methods

### 4.1. Cell Isolations, Cultures and Stimulations

Mouse cells were obtained as previously described [45]. Total-cell populations from spleen and LNC and magnetically purified spleen CD4^+^ cells were stimulated in vitro (1 × 10^6^ cells/mL) by using coated-plate anti-CD3 antibody (clone 145-2C11 eBioscience, 10 μg/mL) and soluble anti-CD28 (clone 37.51 eBioscience, San Diego, CA, USA, 0.5 μg/mL; αCD3/αCD28) in Click’s Medium [46]. After three-day culture, supernatants were collected for cytokine quantification. For RNA analysis, cells were harvested after 24 h of stimulation. When a second round of TCR stimulation was required, cells were washed and re-stimulated by using αCD3/αCD28 at 1 × 10^6^ cells/mL. Specific culture conditions for Th1- or Th17-biased phenotypes were carried out in the presence of IL-12 or IL6 plus TGFβ, respectively, as described [47]. For CD11b^+^ cell depletion, anti-CD11b antibody (Mac-1α, Miltenyi Biotec, Bergisch Gladbach, Germany) was used following the manufacturer’s instructions. Peritoneal macrophage recruitment was induced in mice by using intraperitoneal injections of 2.5 mL of 3% thioglycollate broth (Difco). After four days, the peritoneal fluid was aspirated and cells were harvested, and, after washing them with PBS, seeded in DMEM (10% FCS). Adhesion to the plastic wells was allowed for 3–5 h. Afterwards, non-adherent cells were washed away and adherent macrophages were cultured for 12–16 h in DMEM (0.5% FCS). For the polarisation to M1 and M2 phenotypes, the macrophages were cultured in the presence of LPS (100 ng/mL) or the combination of IL-4, IL-10 and TGFβ (20 ng/mL each), respectively. Cells were harvested for RNA extractions at the indicated times in each case.

### 4.2. ELISA Determinations

Quantification of cytokines released into the extracellular medium was carried out by using enzyme-linked immunosorbent assay (ELISA). Pairs of detection/capture antibodies were: 11B11(ATCC HB188)/BVD6-24G2 (Becton-Dickinson, Franklin Lakes, NJ, USA) for IL-4, Jes5-2A5/Jes5-16E3 (BD-Bioscience) for IL-10, R4-6A.2 (eBioscience, San Diego, CA, USA)/XMG1.2 (Becton-Dickinson) for IFNγ and eBio17CK15A5/eBio17B7 (eBioscience) for IL-17. The label on the detection antibodies was biotin. Reference standard curves were established by using serial dilutions of each cytokine (PeproTech, Rocky Hill, NJ, USA) with known concentrations. The followed procedure was as previously described [48].

### 4.3. Quantitative Real-Time RT-PCR (RT-qPCR)

RNA was extracted using the RNeasy kit (Qiagen, Hilden, Germany) following the manufacturer’s recommendations. Remaining DNA was removed by RNase-free DNase (Qiagen). RNA was retro-transcribed to cDNA by using M-MLV reverse transcriptase (RT; M1705 Promega, Madison, WI, USA) with poly-dT as primer. Real-time PCR was carried out by using GoTaq (Promega, A6002) with SYBR Green in LightCycler equipment (Roche, Holding AG, Basel, Switzerland). The primer sets used for each murine gene are detailed in Table 1.

PCR product quality was checked by using a melting curve analysis for each sample, and the reaction efficiencies were checked to be near 2. Analysis of data for quantification of gene expressions was performed as described [49]. The housekeeping β-actin gene was used to normalise data.

### 4.4. Animals, EAE Inductions, Clinical Evaluations and Treatments

Heterozygous Ptch1tm1Mps mice generated by Dr. Matthew Scott, Stanford University School of Medicine, were obtained from the Jackson Laboratory (Ptch1tm1Mps/J Stock No. 003081) and bred in C57BL/6NCrl strain. wt littermates were used as controls. All experiments were conducted under institutional ethical and safety guidelines with approval number 017/15 of the Madrid Regional Authority’s Ethics Committee in accordance with European Union legislation. EAE was induced in mice 10 to 14 weeks old by using subcutaneous injections of myelin oligodendrocyte glycoprotein (MOG)_35–55_ peptide (Peptide 2.0, Chantilly, VA, USA). The followed procedure was as previously described [45]. Clinical signs were scored on the following scale: no clinical signs, 0; loss of tail tonicity, 1; rear limb weakness, 2; paralysis of one rear limb, 3; paralysis of two rear limbs, 4; full paralysis of four limbs, 5. At value 4, animals were sacrificed to avoid further progression of the disease. Daily score values shown were the averages of those assigned to each mouse by at least two independent observers in blind inspection.

### 4.5. Histopathology

Mice were anesthetised by using intraperitoneal administration of Ketamine-Xylazine and transcardially perfused with PBS. Spinal cords were fixed in 10% formalin solution, processed following standard procedures and paraffin embedded, and 2.5 μm sections were stained with H and E for histological analyses. For immuno-histochemical analyses, slides containing paraffin sections were deparaffinised and boiled in a microwave oven in 10 mM sodium-citrate buffer, peroxidase activity was inhibited with 2% hydrogen peroxide and slides were incubated overnight with primary antibodies, followed by 1 h incubation with appropriate secondary antibodies and 30 min incubation in Vectastain ABC reagent (Vector Laboratories, Burlingame, CA, USA) and developed in DAB solution (Vector Laboratories). GFAP/phospho-Stat3 and GFAP/Olig2 double immunohistochemistry was performed using a BOND RX fully automated research stainer (Leica Biosystems, Wetzlar, Germany) using the manufacturer’s reagents and protocols. Primary antibodies used for immunohistochemistry analysis were CD11b rabbit monoclonal (Abcam, Cambridge, United Kingdom, Ab133357), GFAP rabbit monoclonal (MilliporeSigma, Burlington, MA, USA, 04-1062), Phospho Stat3 (Tyr705) rabbit polyclonal (Elabscience E-AB-20983), β-galactosidase rabbit polyclonal (Cappel, MP Biomedicals, Fountain Parkway Solon, Ohio, USA, 55976), NG2 rabbit polyclonal (Abcam ab101807), human/mouse, Olig1 mouse monoclonal (R&D systems, Minneapolis, MN, USA, MAB2417), Olig2 Rabbit monoclonal (EPR2673) and Iba-1 goat polyclonal (Abcam Ab5076). Secondary antibodies used in immunohistochemistry were donkey anti-rabbit biotin and donkey anti-goat biotin from Jackson ImmunoResearch. Secondary antibodies used in immunofluorescences were 488 and 594 Alexa Fluor conjugated donkey anti-rabbit, donkey anti-goat and donkey anti-mouse antibodies.

### 4.6. Statistics

Statistical analyses were performed with Graph Pad Prism 9.0.2. The tests used to analyse the group differences were unpaired *t*-test for cytokine levels, paired Wilcoxon test for EAE clinical signs and Chi-square test for disease incidence. The number of samples for cytokine determinations and for clinical scores are indicated in each figure legend. Statistical significance is indicated as * *p* < 0.05, ** *p* < 0.01, *** *p* < 0.001, **** *p* < 0.0001 and ns (not significant).

## Figures and Tables

**Figure 1 ijms-23-03171-f001:**
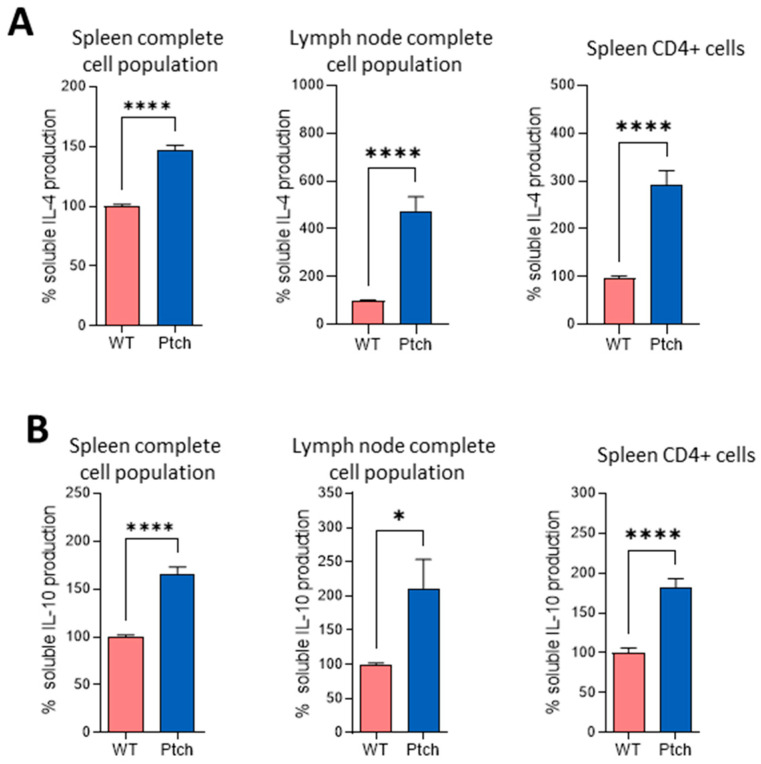
Th2 cytokines produced by T lymphocytes from *ptch*^+/−^ and wt mice. Total-cell populations from spleen or lymph node (LN) or purified spleen CD4^+^ cells from *ptch*^+/−^ and wt mice were stimulated in vitro by using αCD3/αCD28. Three-day culture supernatants were subjected to ELISA analysis for IL-4 (**A**) and IL-10 (**B**) determinations. The results are represented as percentages of the values found for wt mice. Data are the mean of six independent experiments in which each sample was assayed in quadruplicate. Each sample corresponds to a pool of spleen or LN cells from four mice. Standard errors of the means are shown. Statistical significance is indicated as * *p* < 0.05 and **** *p* < 0.0001.

**Figure 2 ijms-23-03171-f002:**
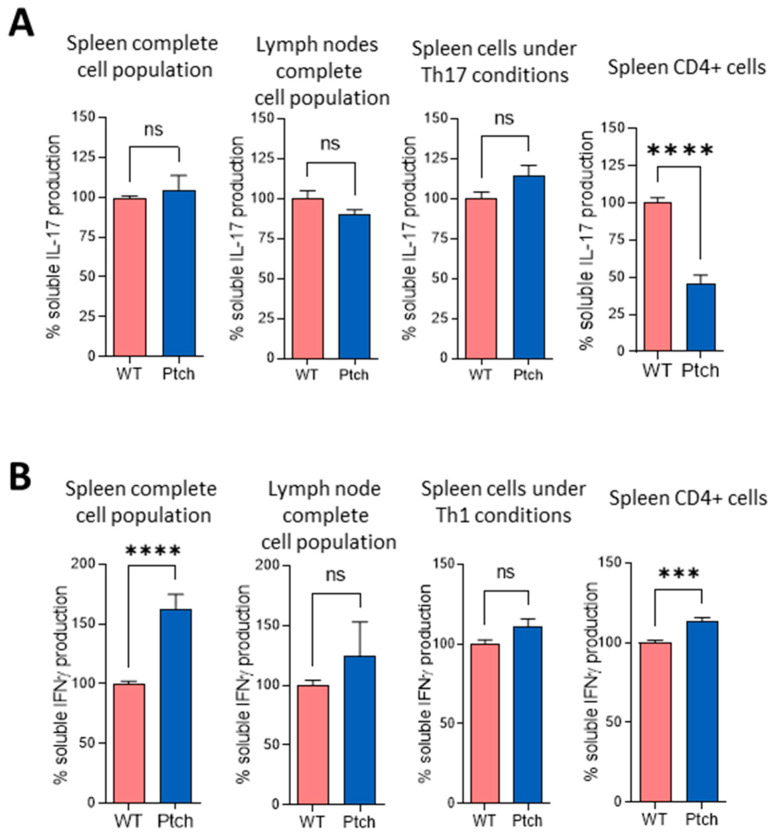
Th1 and Th17 cytokines produced by T lymphocytes from *ptch*^+/−^ and wt mice. Total-cell populations from spleen or LN or purified spleen CD4^+^ cells from *ptch*^+/−^ and wt mice were stimulated in vitro by using αCD3/αCD28. Cells were cultured under non-polarising conditions or under T helper (Th) 17 (**A**) or Th1 (**B**) polarising conditions where indicated. Three-day culture supernatants were subjected to ELISA analysis for IL-17 (**A**) and IFNγ (**B**) determinations. The results are represented as percentages of the values found for wt mice. Data are the mean of five independent experiments in which each sample was assayed in quadruplicate. Each sample corresponds to a pool of spleen or LN cells from four mice. Standard errors of the means are shown. Statistical significance is indicated as *** *p* < 0.001, **** *p* < 0.0001 and ns (not significant).

**Figure 3 ijms-23-03171-f003:**
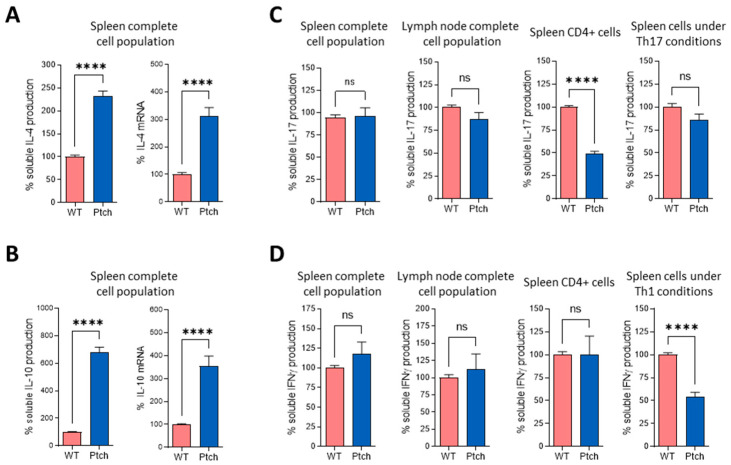
Cytokine profile of T lymphocytes from *ptch*^+/−^ and wt mice after a second round of TCR stimulation. Total-cell populations from spleen or LN or purified spleen CD4^+^ cells from *ptch*^+/−^ and wt mice were stimulated in vitro by using αCD3/αCD28. After three days of culture, cells were subjected to a second round of αCD3/αCD28 for another three days. Soluble IL-4 (**A**), IL-10 (**B**), IFNγ (**C**) and IL-17 (**D**) in the supernatants were measured by using ELISA. IL-4 (**A**) and IL-10 (**B**) mRNA were quantified by using RT-PCR. The results are represented as percentages of the values found for wt mice. Data are the mean of four independent experiments in which each sample was assayed in quadruplicate. Each sample corresponds to a pool of spleen or LN cells from four mice. Standard errors of the means are shown. Statistical significance is indicated as **** *p* < 0.0001 and ns (not significant).

**Figure 4 ijms-23-03171-f004:**
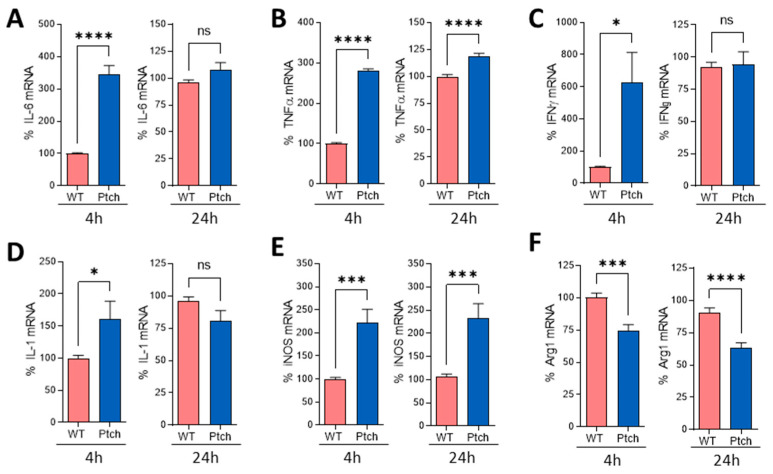
M1/M2 markers of macrophages from *ptch*^+/−^ and wt mice. Peritoneal macrophages were cultured in M1 (LPS, 100 ng/mL) (**A**–**E**) or M2 (IL-4, IL-10 and TGFβ, 20 ng/mL each) (**F**) conditions. At 4 or 24 h (indicated), cells were subjected to RNA extraction and the gene expressions of the gene products specified in each panel were quantified by using RT-PCR. The results are represented as percentages of the values found for wt mice. Data are the mean of three independent experiments in which each sample was assayed in triplicate. Each sample corresponds to a pool of macrophages from three mice. Statistical significance is indicated as * *p* < 0.05, *** *p* < 0.001, **** *p* < 0.0001 and ns (not significant). Standard errors of the means are shown.

**Figure 5 ijms-23-03171-f005:**
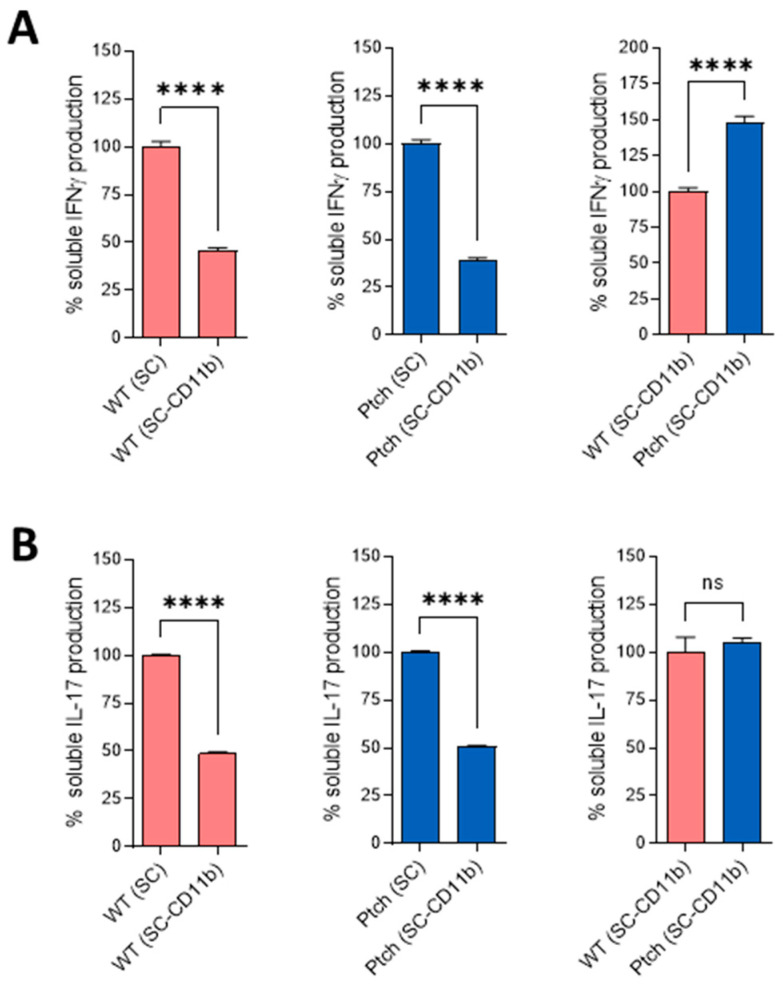
Effect of CD11b^+^ fraction depletion in complete spleen-cell populations. Spleen cells from wt and *ptch*^+/−^ mice were depleted of the CD11b fraction and stimulated by using αCD3/αCD28 for three days. The IFNγ (**A**) and IL17 (**B**) productions were determined by using ELISA. The different compared populations are specified at the bottom of each histogram. Data are the means of three independent experiments in which each sample was assayed in quadruplicate. Each sample corresponds to a pool of spleen or LN cells from three mice. Statistical significance is indicated as **** *p* < 0.0001 and ns (not significant). Standard errors of the means are shown.

**Figure 6 ijms-23-03171-f006:**
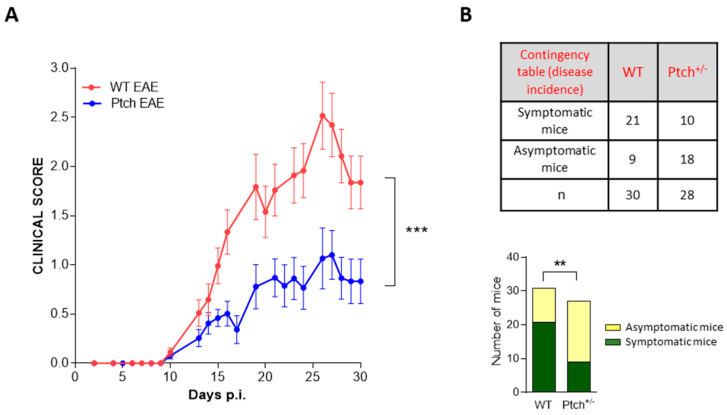
Progression of EAE in *ptch*^+/−^ and wt mice. Groups of 12–16 mice were induced to EAE by using MOG_35–55_ peptide inoculation. (**A**): Daily clinical score. Values were obtained from the mean of two experiments. (**B**): Disease incidence contingency table. Each individual mouse was considered symptomatic for disease onset data only if it maintained a score greater than one for at least three days. The bar graphs for representation of the disease incidence contingency table show the number of symptomatic and asymptomatic mice at day 30 p.i. for each group. The n values are indicated on the table. Statistical significance is indicated as ** *p* < 0.01 and *** *p* < 0.001. Standard errors of the means are shown.

**Figure 7 ijms-23-03171-f007:**
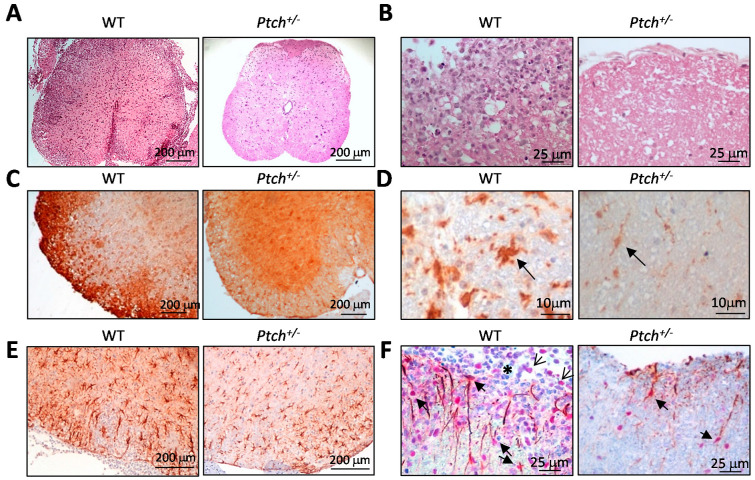
Histological and immuno-histochemical analyses of spinal cord cross-sections from *ptch*^+/−^ and wt mice. MOG_35–55_-inoculated mice were sacrificed on day 20 after inoculation, and spinal cord sections were subjected to haematoxilin-eosin staining (**A**,**B**) or immuno-histochemical analysis with antibodies against CD11b (**C**), Iba-1 (**D**) and GFAP (**E**) or double staining for GFAP and Stat3 (**F**). Arrows in D point to microglia with typical activated or resting morphologies in wt or *ptch*^+/−^ samples, respectively. Thick arrows in F indicate double-positive cells with anti-GFAP (brown cytoplasm) and anti-Stat3 (red nuclei), thin arrows show infiltrating immune cells and the asterisk marks polymorphic leukocytes. Magnitude scales are indicated on each micrograph.

**Figure 8 ijms-23-03171-f008:**
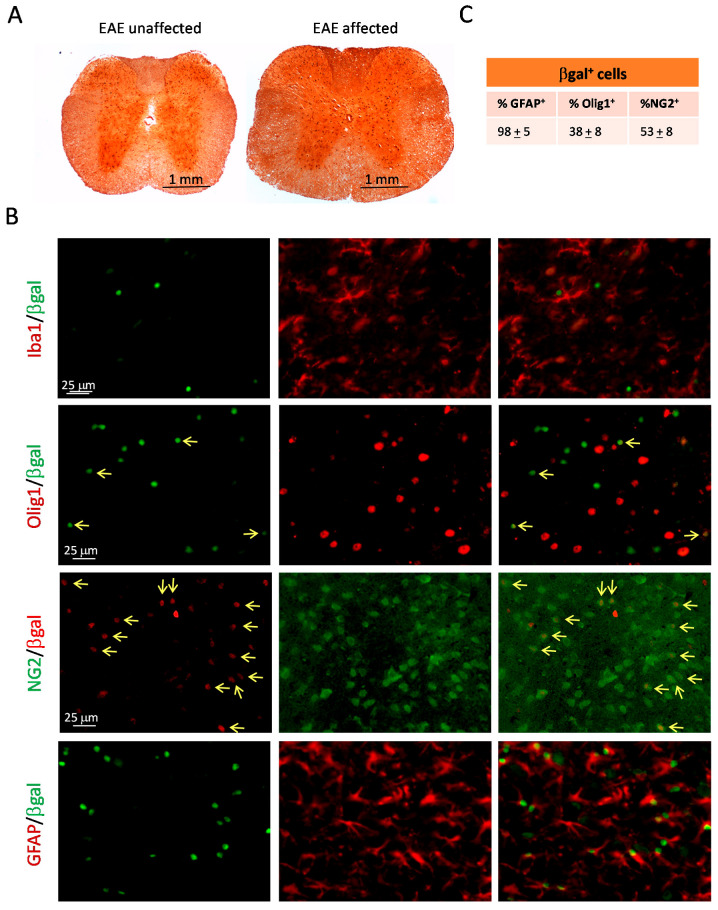
Detection of β-gal-positive cells in *ptch*^+/−^ mice. (**A**): Immuno-histochemistries by using anti-βgal antibody in cross-sections of spinal cords from *ptch*^+/−^ mice inoculated with MOG_35–55_ after 20 days from inoculation. Samples from mice with different clinical scores are shown (symptomatic, clinical score: 4 or asymptomatic, clinical score: 0). (**B**): Double immunofluorescence analyses of spinal cords from symptomatic *ptch*^+/−^ mice for β-gal and Iba1 (first row, green nuclei and red cytoplasm), Olig1 (second row, green or red nuclei), NG2 (third row, red nuclei and green cytoplasm) or GFAP (fourth row, green nuclei and red cytoplasm). Last column shows the merged images; yellow arrows point to double-positive cells. (**C**): Percentages of βgal^+^ cells co-expressing GFAP, Olig1 or NG2.

**Table 1 ijms-23-03171-t001:** Sequences of the primer sets used for real time PCR.

	Forward (5’-3’)	Reverse (5’-3’)
IL-4	ATCCTGCTCTTCTTTCTCG	GATGCTCTTTAGGCTTTCC
IL-10	GCCTTATCGGAAATGATCCA	GCTCCACTGCCTTGCTCTTA
IFNγ	TGCATCTTGGCTTTGCAGCTCTTCCTCATGGC	TGGACCTGTGGGTTGTTGACCTCAAACTTGGC
IL-17	CGCAAAAGTGAGCTCCAGAA	CCTCTTCAGGACCAGGATCT
IL-6	GCCTTCTTGGGACTGATGCT	GACAGGTCTGTTGGGAGTGG
IL-1β	AGAGCCCATCCTCTGTGACT	GGAGCCTGTAGTGCAGTTGT
TNFα	CGTCAGCCGATTTGCTATCT	CGGACTCCGCAAAGTCTAAG
iNOS	CTCACTGGGACAGCACAGAA	GCTTGTCTCTGGGTCCTCTG
Arg1	CATGGAAGTGAACCCAACTCTTGG	TCAGTCCCTGGCTTATGGTTACCC
β-actin	TGTTACCAACTGGGACGACA	GGGGTGTTGAAGGTCTCAAA

## Data Availability

Not applicable.

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
