# Peer review of "Hedgehog Signalling Modulates Immune Response and Protects against Experimental Autoimmune Encephalomyelitis"

_ijms, 2022, doi:10.3390/ijms23063171_

Round 1

Reviewer 1 Report

The manuscript is scientifically sound, well written, the data are properly analysed and the results are well presented. This study could be of interest to many researchers in the field.

Author Response

We would like to thank Reviewer 1 for his positive view of our manuscript and the time spent reviewing it.

Reviewer 2 Report

the authors of the manuscript " Hedgehog signalling modulates.......Encephalomyelitis" has been very well designed and executed and excellent written.  The affect of partial loss of Hh in modulating the the immune response very well done. 

there is only 2 questions:

  1. Why the authors used peritoneal macrophage in experimental  Encephalomyelitis.
  2. Macrophages grown in presence of LPS will screw the macrophages to M1 so why the authors used LPS induced M1 macrophages and what did the authors use to screw the macrophages to M2.

Author Response

We would like to thank reviewer 2 for his comments and for giving us the chance to clarify his doubts regarding the macrophage experiment in Figure 4.

  1. We must clarify that in the experiment in Figure 4, where the response of peritoneal macrophages to stimulation towards phenotypes M1 or M2 is analyzed, there is no induction to EAE, since the objective of this experiment was to compare macrophage polarization in ptch+/- versus wt animals in the absence of pathology.
  2. Culture conditions for phenotype polarizations to M1 and M2 are described in the Materials and Methods section. However, we have added them in the legend of Figure 4 to make it easier to find them for the reader (M1: 100 ng/ml of LPS; M2: IL-4, IL-10 and TGFb, 20 ng/ml each).

Reviewer 3 Report

This is very interesting and good paper. I have only few comments.

  1. L452- Please make table with primer list.
  2. Please indicate how many samples you have in each experiment in figure legend and section 4.6.
  3. Reference section should be amened according to journal's policy.
  4. Fig. 8B should be quantitatively scored by preparing additional plots (like bottom panel).
  5. Please indicae statistical value level of *, **, *** in each figure legend.
  6. some typo erros in the manuscript should be amended (eg., L153-154,

Author Response

We would like to thank the reviewer for his constructive comments. Below we present the answers to each of them:

  1. The primer list has been replaced by a table as suggested by the reviewer.
  2. The number of samples is now indicated in each figure legend. For spleen cells, LN cells and peritoneal macrophages, each sample correspond to a pool of cells obtained from 3-4 mice. Each sample was assayed in triplicate or quadruplicate (specified in each figure legend), and data shown are the mean of 3-6 independent experiments (also specified in each figure legend) represented as percentages of the values found for wt mice. For clinical score of EAE, the data were obtained from two independent experiments in which groups of 12-16 mice were analyzed in each.
  3. The references format has been corrected according to the IJMS instructions.
  4. We think that this observation is due to a formal error that we had overlooked in our first version: Figure 8 lacked labels for panels A, B and C. In addition, to try to maintain the geometrical shape of the figure, panel C is placed on top of panel B. We believe that, due to the lack of labels, reading became difficult and could have led to misinterpretation. We have corrected this deficiency in the new version of our manuscript by adding its label to each of the panels.
  5. The statistical value level is indicated in the Materials and Methods section. However, we have added this information in each figure legend to make it easier to understand each Figure (*p < 0.05, **p < 0.01, ***p < 0.001, ****p < 0.0001, ns: not significant)
  6. We have carefully reviewed and corrected the typo errors throughout the manuscript.